# Molecular Evaluation of the Impact of Polymorphic Variants in Apoptotic (*Bcl-2/Bax*) and Proinflammatory Cytokine (*TNF-α/IL-8*) Genes on the Susceptibility and Progression of Myeloproliferative Neoplasms: A Case-Control Biomarker Study

**Mamdoh S. Moawadh** [1], **Rashid Mir** [1,2,*], **Faris J. Tayeb** [1,3], **Orooba Asim** [2,†] and **Mohammad Fahad Ullah** [1]

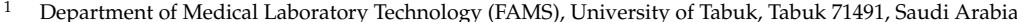

[1] Department of Medical Laboratory Technology (FAMS), University of Tabuk, Tabuk 71491, Saudi Arabia
[2] Division of Molecular Biology, Prince Fahd Chair for Biomedical Research, University of Tabuk, Tabuk 71491, Saudi Arabia
[3] Community College, University of Tabuk, Tabuk 71491, Saudi Arabia
[*] Correspondence: rashid@ut.edu.sa; Tel.: +966-580225811
[†] Internship Trainee.

**Abstract:** The regulation of apoptosis (the programmed cell death) is dependent on the crucial involvement of BCL2 and BAX. The *Bax-248G>A* and *Bcl-2-938 C>A* polymorphic variations in the promoter sequences of the *Bax* and *Bcl-2* gene have been recently associated with low *Bax* expression, progression to advanced stages, treatment resistance, and shortened overall survival rate in some hematological malignancies, including chronic myeloid leukemia (CML) and other myeloproliferative neoplasms. Chronic inflammation has been linked to various stages of carcinogenesis wherein pro-inflammatory cytokines play diverse roles in influencing cancer microenvironment leading to cell invasion and cancer progression. Cytokines such as TNF-α and IL-8 have been implicated in cancer growth in both solid and hematological malignancies with studies showing their elevated levels in patients. Genomic approaches have in recent years provided significant knowledge with the regard to the association of certain SNPs (single nucleotide polymerphisms) either in a gene or its promoter that can influence its expression, with the risk and susceptibility to human diseases including cancer. This study has investigated the consequences of promoter SNPs in apoptosis genes *Bax-248G>A (rs4645878)*/*Bcl-2-938C>A (rs2279115)* and pro-inflammatory cytokines *TNF-α rs1800629 G>A/IL-8 rs4073 T>A* on the risk and susceptibility towards hematological cancers. The study design has 235 individuals both male and female enrolled as subjects that had 113 cases of MPDs (myeloproliferative disorders) and 122 healthy individuals as controls. The genotyping studies were conducted through ARMS PCR (amplification-refractory mutation system PCR). The *Bcl-2-938 C>A* polymorphism showed up in 22% of patients in the study, while it was observed in only 10% of normal controls. This difference in genotype and allele frequency between the two groups was significant ($p = 0.025$). Similarly, the *Bax-248G>A* polymorphism was detected in 6.48% of the patients and 4.54% of the normal controls, with a significant difference in genotype and allele frequency between the groups ($p = 0.048$). The results suggest that the *Bcl-2-938 C>A* variant is linked to an elevated risk of MPDs in the codominant, dominant, and recessive inheritance models. Moreover, the study indicated allele A as risk allele which can significantly increase the risk of MPDs unlike the C allele. In case of *Bax* gene covariants, these were associated with an increased risk of MPDs in the codominant inheritance model and dominant inheritance model. It was found that the allele A significantly enhanced the risk of MPDs unlike the G allele. The frequencies of *IL-8 rs4073 T>A* in patients was found to be TT (16.39%), AT (36.88%) and AA (46.72%), compared to controls who were more likely to have frequencies of TT (39.34%), AT (37.70%) and AA (22.95%) as such, respectively. There was a notable overrepresentation of the AA genotype and GG homozygotes among patients compared to controls in *TNF-α* polymorphic variants, with 6.55% of patients having the AA genotype and 84% of patients being GG homozygotes, compared to 1.63% and 69%, respectively in controls. The data from the current study provide partial but important evidence that polymorphisms in

apoptotic genes *Bcl-2-938C>A* and *Bax-248G>A* and pro-inflammatory cytokines *IL-8 rs4073 T>A* and *TNF-α G>A* may help predict the clinical outcomes of patients and determine the significance of such polymorphic variations in the risk of myeloproliferative diseases and their role as prognostic markers in disease management using a case-control study approach.

**Keywords:** single nucleotide polymorphisms; apoptosis; gene promoter; proinflammatory cytokines; MPDs; CML—chronic myeloid leukaemia; PV—polycythemia vera; ET—essential thrombocythemia

## 1. Introduction

The statistical figures show that blood cancers or hematological malignancies have a high incidence rate worldwide. In Western societies, leukemia is considered to represent a high incidence of blood cancer, affecting 4.2 per hundred thousand annually and globally, it represents 2.8% of all new cancer cases (300,000 cases/year) [1,2]. Evading apoptosis is the mainstay signature in blood cancers and also the modulation of proinflammatory cytokines that act as mediators of cancer microenvironment and cancer progression. The mechanism of apoptosis defines a committed process of cell death that is used as a termination stage of the cell fate during certain damaging situations such as infections, DNA damage, or any morphological or phenotypic characteristic that might be harmful if the cell survives and proliferates [3,4]. When the cell cycle is abnormally interrupted, apoptosis may be evaded causing failure of the preventive responses resulting in disorders including cancers, autoimmune diseases, degenerative diseases, and psychiatric illnesses [5,6]. BCL-2 and BAX include the *Bcl-2* family of protein members, which play a crucial role in the mechanisms related to mitochondrial dysfunction and programmed cell death [7,8]. Apoptosis is a delegated and sensitive process which primes via two distinct pathways: death receptor signaling and mitochondrial (BCL2-dependant) pathways [9]. In the mitochondrial cascade, over 20 BCL2 family proteins, including BCL-2 which is an anti-apoptotic protein have been shown to be involved in cell signaling pathways related to apoptosis. The *Bcl-2* gene has been demonstrated to be highly expressed in blood cancers, and the level of *Bcl-2* expression has been found to be elevated in the accelerated/blastic phase of CML cells compared to the chronic stage which might drive the progression to the advanced stages [10]. The *Bcl-2* gene contains two promoters: P1 and P2. Some studies have reported that in case of P2, a pathogenic SNP (−938 C>A) correlates with altered expressions of the BCL-2 protein. The BCL-2 protein is thus crucial to apoptosis, and as such certain genotypes due to polymorphic variations may affect cell-death intrinsic pathways in a multitude of disorders [11–14]. It is understood that certain SNPs have the ability to influence disease progression and may also result in resistance to effective treatment. Interestingly, some other studies suggested the positive association for better overall survival of patients owing to the presence of certain *Bcl-2* SNPs [15–17]. The BAX protein is encoded with a promoter and six exons [18]. The potential of this promoter lies in its ability to interact with a range of transcription factors and regulator proteins, such as the NF-κB binding site, p53 response element, the TATA box and the canonical E-box [18]. Several studies have found that *Bax-248G>A* genotype is linked with the risk of human cancer [19]. The BCL2 family proteins utilize BAX as an anti-apoptotic regulator and employ the pro-apoptotic BCL-2 proteins to disrupt the balance between cell death and survival in various cell death pathways. These proteins act as primary intracellular effectors [20,21]. SNPs related to *Bcl-2/Bax* gene promoters have been investigated in several studies on cancer initiation, development and progression. It has been reported that some SNPs such as *Bax-248G>A(rs4645878)* and (or) *Bcl-2-938C>A (rs2279115)* show alterations in the gene expression in a pathogenic-manner, for example, those found in acute lymphoblastic leukemia [14] and certain solid tumors [22,23]. The overexpression of *Bcl-2* and underexpression of *Bax*, resulting from the genetic variants of the encoded genes, have been linked to the development of these cancers [24,25].

Chronic inflammation has been linked to various stages of carcinogenesis wherein pro-inflammatory cytokines have been shown to be expressed in tissues undergoing transformation and advanced stages of cancer including metastasis [26,27]. Several studies have also provided a link between pro-inflammatory cytokines and apoptosis in the pathology of certain diseases [28–30]. Mechanistic studies have shown the modulation of apoptosis related proteins such as BCL-2 and BAX with varying expression of *IL-8* suggesting the involvement of Hsp60-IL-8 axis in apoptosis resistance in cancer [31]. Taken together, these studies provide evidence for the intricate interplay between apoptotic genes and proinflammatory cytokines in cancer, underscoring the necessity for additional research in this field. Cytokines which are functional proteins that have been known to act as the modulator of immune response and inflammation, have assumed novel roles as modulator of carcinogenesis due to their elevated levels which are associated with development and progression of cancer disease [32,33]. As the name suggests, the TNF-$\alpha$ was originally identified as a factor produced from activated T-lymphocytes, macrophages, and natural killer cells which can cause necrosis of the tumor cells; however, it has now been found to have a pleiotropic action mechanism that can influence various signal transduction pathways related to cell survival and death in addition to immunological cascades [34]. In addition to its role in influencing a tumor microenvironment in the case of solid malignancies, TNF-$\alpha$ has been found to be associated with adverse clinical manifestations of leukemia including extramedullary infiltration and resistance to chemotherapy [35]. It has been suggested that polymorphic gene variations in the *TNF-$\alpha$* promoter region that include 308G/A and 238G/A, might possess the risk of different cancers including colorectal cancer by regulating TNF-$\alpha$ cytokine synthesis [36]. Interleukin-8 is another pro-inflammatory cytokine, which is largely acting as a chemo-attractant in inflammation targeting neutrophils to the site of inflammation [37]. A study on patients with acute myeloid leukemia has provided evidence of high *IL-8* expression, which could be correlated with poor prognosis in certain AML subsets based on differences between genetic subgroups, due to its ability to act as mediator to influence cancer microenvironment [38]. Previous research has shown a noteworthy association between the expression of interleukin-8 and its receptor in individuals with myeloid and lymphoid leukemia [39]. Thus, it is evident that cytokines have clinicobiological implications in the induction and progression of hematological malignancies and are potential molecules as disease markers and therapeutic targets [40].

Genomic approaches have in recent years provided significant knowledge with the regard to the association of certain SNPs in either a gene or its promoter that can influence its expression, with the risk and susceptibility to human diseases including cancer [41]. This study has investigated the influence of promoter SNPs in apoptosis genes that included *Bax-248G>A (rs4645878)/ Bcl-2-938C>A (rs2279115)* and pro-inflammatory cytokines including *TNF-$\alpha$ rs1800629 G>A/ IL-8 rs4073 T>A* on hematological cancers that include CML and other MPDs, and hence, determine the significance of such polymorphic variations in the risk of myeloproliferative disorders and their role as prognostic markers in disease management.

## 2. Experimental Methods

### 2.1. Study Population

The cohort study included 235 subjects among which 113 were histologically as well as clinically confirmed cases of myeloproliferative disorders and 122 healthy controls as shown in Figure 1 below. Before sampling, patient consent papers were obtained from each leukemia patient. The patients were assigned from various Tabuk-area hospitals such as King Khaled Hospital, King Fahd Specialist Hospital and Armed Forces Hospital—Tabuk.

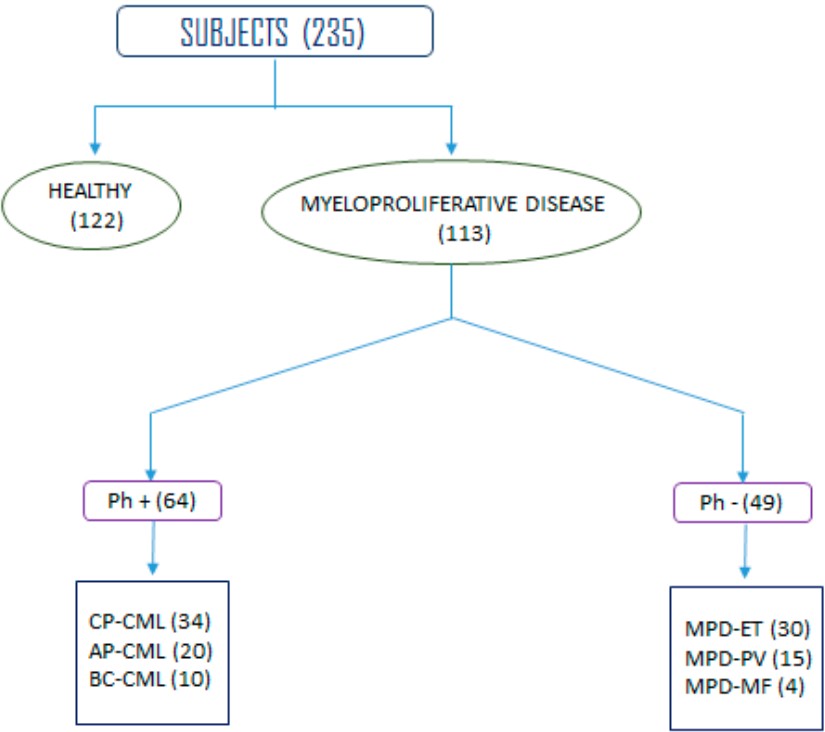

**Figure 1.** Characterization of human subjects enrolled in the study.

## 2.2. Inclusion and Exclusion Criteria

The study included 113 clinically confirmed cases of myeloproliferative disorders. Further validation was performed by bone marrow hyperplasia and associated clinical characteristics that included splenomegaly and hepatomegaly besides other confirmatory clinical findings. The study excluded the individuals (patients) with a history of any previous malignant conditions.

## 2.3. Peripheral Blood Collection from Controls and Patients Cohorts

To collect samples, 3 mL of peripheral blood was taken from all patients and controls in EDTA vials, and informed consent was obtained from each of them. Once the Institutional Ethics Committee approved the study (Approval no. UT-88-19-2019 Dated 24 November 2019), the samples were collected, stored and processed. For the healthy control group, individuals who came in for a routine check-up were selected.

## 2.4. Selection Criteria of Healthy Controls

To establish a group of healthy controls, 122 individuals between the ages of 20 and 50 who were visiting the hospital for routine check-ups were included in the study. The control group was chosen from the general population of the same geographic area. Socio-demographic variables such as age, sex, and lifestyle were recorded using a questionnaire, and regular medical check-ups were conducted, including tests such as complete blood count, kidney function test, liver function test, etc. Additionally, the participants' medical history was also noted. Individuals who did not show any symptoms of illness and had no previous medical history of serious illnesses or chronic conditions were considered to be in good health.

## 2.5. Genomic DNA Extraction

To extract DNA from the peripheral blood, the DNeasy Blood Kit (Cat No. 69506) from Qiagen, Germany was used with the described protocol. The extracted DNA was solubilized in nuclease-free water and placed at 4 °C for the time it had to be used for analysis. The quality of the DNA obtained from the patients and healthy controls was

evaluated through 1% agarose gel electrophoresis, while the quantity of the extracted DNA was estimated using a NanoDrop™ (Thermo Scientific, Waltham, MA, USA) by measuring absorbance at 260 nm and 280 nm.

### 2.6. Genotyping of Bcl-2-938 C>A, Bax G>A, TNF-α G>A and IL-8 T>A

The genotyping of SNPs for *Bcl-2 rs2279115C>T*, *Bax−248G>A* rs4645878, *TNF-α rs1800629 G>A* and *IL-8 rs4073 T>A* was performed using amplification-refractory mutation system PCR (ARMS-PCR). The *Bax G (−248) A* rs4645878 SNP ARMS primers were designed by using Primer 3 software. For *Bcl-2 rs2279115C>T*, ARMS primers were previously used by Butt et al. [42], for *TNF-α rs1800629 G>A* as well as for *IL-8 rs4073 T>A* SNPs, ARMS primers were previously used by Matheson et. al. [43]. The primer sequences have been listed in Table 1:

**Table 1.** ARMS primers for Genotyping.

| Direction | ARMS Primer Sequence | Annealing Tempt | Product Size |
|---|---|---|---|
| *Bax G(−248) A rs4645878* gene polymorphism | | | |
| *Bax FO* | 5′-CCT GGA AGC ATG CTA TTT TGGGCCT-3′ | 65 °C | 323 bp |
| *Bax RO* | 5′-ACG TGA GAG CCC CGC TGA ACG T-3′ | | |
| *Bax FI-G* | 5′-GGC ATT AGA GCT GCG ATT GGA CTG G-3′ | | 209 bp |
| *Bax RI-A* | 5′-AGTGGCGCCGTCCAACAGCAGT-3′ | | 160 bp |
| *Bcl-2-938 C>A rs2279115* gene polymorphism | | | |
| *Bcl-2-FO* | 5′-CCG GCT CCT TCA TCG TCT CC-3′ | 58 °C | 300 bp |
| *Bcl-2-RO* | 5′-CCC AGG AGA GAG ACA GGG GAA AT-3′ | | |
| *Bcl-2-FI-A* | 5′-AATAAAACCCTCCCCCACCACCT-3′ | | 220 bp |
| *Bcl-2-RI-C* | 5′-CCCTTCTCGGCAATTTACACGC-3′ | | 121 bp |
| *TNF-α rs1800629 G>A* gene polymorphism | | | |
| *TNF-αF0* | 5′-ACC CAA ACA CAG GCC TCA GGACTCAACA-3′ | 62 °C | 323 bp |
| *TNF-αR0* | 5′-TGGAGGCAATAGCTTTTGAGGGGCAGGA-3′ | | |
| *TNF-αFI-A* | 5′-AGTTGGGGACACGCAAGCATGAAGGATA-3′ | | 154 bp |
| *TNF-α RI-C* | 5′-TAGGACCCTGGAGGCTAGACCCCGTACC-3′ | | 224 bp |
| *IL8-251T>A (rs4073)* gene polymorphism | | | |
| *IL8-Fo* | 5′-CAT GAT AGC ATC TGT AAT TAA CTG-3′ | 58 °C | 349 bp |
| *IL8-Ro* | 5′-CACAATTTGGTGAATTATCAAA-3′ | | |
| *IL8-FI-A* | 5′-GTTATCTAGAAATAAAAAAGCATACAA-3′ | | 228 bp |
| *IL8-RI-T* | 5′-CTCATCTTTTCATTATGTCAGA-3′ | | 169 bp |

Fo—outer forward primer; FI—Inner forward primer; Ro—outer reverse primer; RI—outer reverse primer.

### 2.7. PCR Cocktail and Optimization Procedure by a Gradient PCR

In the ARMS-PCR, a final reaction volume of 25 μL was utilized. This included 50 ng of template DNA, Forward (Outer)-0.30 μL, Reverse (Outer)-0.30 μL, Forward (Inner)-0.20 μL, Reverse (Inner)-0.20 μL of 25 pmol each primer, and 10 μL of GoTaq® Green Master Mix from Promega, Fitchburg, WI, USA (cat no M7122). The reaction mixture also received 2 μL of DNA from either the patient or control sample, and nuclease-free ddH2O was added to adjust the final volume of the PCR tube to 25 μL. Data analysis determined that an optimal temperature of 58 °C was required for the *Bcl-2-938 C>A rs2279115* SNP, which was tested with a gradient PCR thermocycler within a temperature range of 55 °C to 62 °C. Increasing the cycle number from 30 to 40 cycles significantly enhanced the yields of all three PCR products. Similarly, the optimal temperature for the Bax G (−248) A SNP was determined to be 68 °C, selected from the temperature range of 55 °C to 65 °C. These modifications, combined with a less competitive reaction from the control, demonstrate strong amplification of the mutant allele, as indicated by the intensity of the corresponding bands observed during agarose gel electrophoresis.

### 2.8. Thermocycling Conditions and Gel Electrophoresis

The amplification process involved an initial denaturation step at 95 °C for the first 10 min, followed by 45 cycles of amplification, consisting of 35 s at 95 °C, 40 s at 58 °C, and 45 s at 72 °C. This was then followed by a final extension step at 72 °C for 10 min.

The PCR products after amplification were resolved through gel electrophoresis on a 2% agarose gel and observed using a UV transilluminator, Bio-Rad, Hercules, CA, USA with syber safe stain.

*2.9. Statistical Analysis*

To distinguish between the different groups, student's two-sample t-test or one-way analysis of variance were employed for continuous variables, while the χ2 test was utilized for categorical variables such as deviations from Hardy–Weinberg disequilibrium. Furthermore, to evaluate the allelic and genotypic frequencies between the *Bcl-2* and *Bax* gene groups, a Chi-square test was used. The potential relationships between the *Bcl-2-938 C>A rs2279115* SNP and *Bax G (−248) A* SNP, *TNF-α rs1800629 G>A* and *IL-8 rs4073 T>A* genotypes and myeloproliferative leukaemia cases were assessed by estimating odds ratios, risk ratios and risk differences with 95% confidence intervals. A *p*-value of less than 0.05 was considered to be of significance. The analyses were performed using SPSS (Chicago, IL, USA) and Graph Pad Prism 8.4 software (San Diego, CA, USA).

## 3. Results

*3.1. Demographic Characteristics of Study Population*

The data regarding the demographics of the patients and controls are presented in Table 2. Of the study subjects, 82 (72.56%) were male patients, while 31 (27.44%) were female patients. Conversely, among the controls, 74 (60.65%) were males and 48 (39.34%) were females. The majority of the patient group (76, or 67.21%) were over the age of 40, while 37 (32.78%) were under 40. Among the 64 CML cases, 34 (53.12%) were in the early stage (chronic phases), 20 (31.25%) were in the accelerated phase, and 10 (15.62%) were in blast crisis. Additionally, of the patients with other myeloproliferative diseases, 30 had ET, 15 had PV, and 4 had myelofibrosis.

**Table 2.** Characterization of the demographic features of the study population.

|  | Patients (113) | Controls (122) |
|---|---|---|
| Age | N = % | N = % |
| >40 | 76 (67.21%) | 80 (65.57%) |
| <40 | 37 (32.78%) | 42 (34.42%) |
| Gender | N = % | N = % |
| Males | 82 (72.56%) | 74 (60.65%) |
| Females | 31 (27.44%) | 48 (39.34%) |
| Frequency of different stages of CML (Ph+) | | |
|  | | N = 64 |
| Chronic phase CML | | 34 (53.12%) |
| Accelerated phase CML | | 20 (31.25%) |
| Blast crisis CML | | 10 (15.62%) |
| Frequency of other MPDs (Ph−) | | |
| MPD TYPE | | N = 49 |
| Essential thrombocytopenia | | 30 (61.22%) |
| Polycythemia verra | | 15 (30.61%) |
| Myelofibrosis | | 04 (8.16%) |

*3.2. Genotyping and Gel Electrophoresis for the Identification of Alleles*

Genotyping for the four tested genes was performed by using the amplification-refractory mutation system PCR (ARMS-PCR). Finally, the PCR products generated as a result of PCR amplification were resolved by agarose gel electrophoresis (2.5%) and the corresponding bands were identified for wild type or mutant allele as shown in Figure 2.

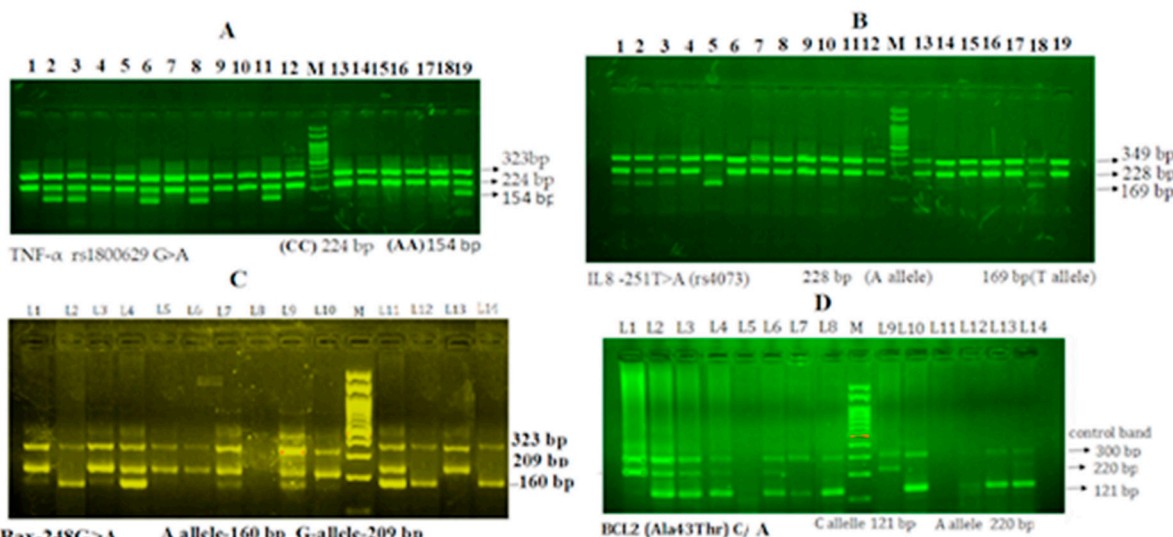

**Figure 2.** Gel electrophoresis of *Bcl-2-938 C>A rs2279115* SNP and *Bax G (−248) A* SNP, *TNF-α rs1800629 G>A* and *IL-8 rs4073 T>A* gene polymorphisms. (**A**) *TNF-α*: M-100 bp DNA ladder, Heterozygous-CA-P2,P3,P6,P8,P1,P18 and homozygous AA-0 and homozygous CC-P1,P4,P5,P6,P8 to P10 and P12 and P13-P17. (**B**) *IL-8*: M-100 bp DNA ladder, Heterozygous-TA-P1,P2,P3,P18 and homozygous TT-P5 and homozygous AA-P4, P6 to p17 and P19. (**C**) *Bax*: M-100 bp DNA ladder, Heterozygous-GA-P1,P3,P4,P7,P9,P11 and homozygous AA-P2,P12,P14 and homozygous GG-P5,P6,P10 and P13. (**D**) *Bcl-2:* M-100 bp DNA ladder, Heterozygous-GA-P1,P2,P3,P5,P14 and homozygous CC-P8,P10,P12,P13 and homozygous AA-P1,P9.

### 3.3. Gel Electrophoresis of TNF-α rs1800629 G>A Gene Amplification

The primers Fo and Ro were designed to flank the exon/intron of the TNF-α gene, which resulted in the generation of a 323 bp band. This band served as a control for assessing the integrity of the DNA. Additionally, when primers Fo and RI were used, a 224 bp band corresponding to the G allele was generated. Alternatively, primers FI and RO produced a 154 bp band from the A allele, as demonstrated in Figure 2A.

### 3.4. Gel Electrophoresis of Interleukin-8 rs4073 T>A Gene Amplification

The external primers Fo and Ro were utilized for the amplification of the external region of the IL-8 gene, generating a 349 bp band that served as a control for evaluating the quality of the DNA. Additionally, primers FO and RI produced a 228 bp band corresponding to the A allele, while primers FI and RO resulted in a 169 bp band corresponding to the T allele, as shown in Figure 2B.

### 3.5. Gel Electrophoresis of Bax G (−248) A Gene Amplification

To assess both the quality and quantity of the DNA, primers Fo and Ro of the Bax2 gene generated a 323 bp band. Additionally, primers FI and Ro were used to amplify the wild-type allele (G allele), which produced a 209 bp band. Alternatively, primers Fo and RI were used to produce a 160 bp band from the mutant A allele, as shown in Figure 2C.

### 3.6. Gel Electrophoresis of Bcl-2-938 C>A Gene Amplification

To ensure the quality and quantity of the DNA, primers Fo and Ro of the Bcl-2 gene generated a 300 bp band. Moreover, primers FI and Ro were utilized to amplify the wild-type allele (C allele), resulting in a 121 bp band. On the other hand, primers Fo and RI generated a 220 bp band from the mutant A allele, as demonstrated in Figure 2D.

## 3.7. Comparative Analysis of the Bcl-2-938 C>A rs2279115, Bax-248G>A, TNF-α rs1800629 G>A and Interleukin-8 rs4073 T>A (IL-8-251T>A) Genotypes in Myeloproliferative Leukemia Patients and Healthy Controls

Our results (Table 3) indicate significant differences in the occurrence of *Bcl-2-938 C>A* rs2279115 CC, CA, and AA genotypes between leukaemia patients and healthy controls ($p = 0.025$). Specifically, the frequency of the AA homozygote was higher in patients (22%) than in controls (10%), while the CC homozygote was more frequent in controls (34%) than in patients (49%). We also found that the A allele was more prevalent among patients (44%) than controls (31%), indicating it as a risk allele, while the C allele acted as a protective factor. Additionally, there were notable differences in the frequencies of *Bax-248 G>A* GG, GA, and AA genotypes between the two groups ($p = 0.048$), with the AA genotype being more common in patients (6.48%) and the GG homozygote being more prevalent in controls (41.66%). Furthermore, we observed significant differences in the frequencies of *IL-8 rs4073 T>A* between leukaemia patients and healthy controls ($p < 0.0001$), with patients having higher frequencies of A allele (0.75 vs. 0.61) and AA genotype, while healthy individuals had higher frequencies of the TT genotype and T allele, indicating their protective effect against the disease. Finally, we found a statistically significant difference in the frequencies of *TNF-α rs1800629* GG, GA, and AA genotypes between leukaemia patients and healthy controls ($p = 0.048$), with the AA and GG homozygote being more common among patients (7.96% and 83.18%, respectively). However, the G allele appeared to be a risk allele with higher frequency (fG) reported among patients (0.87) compared to control (0.84), while the A allele seemed to be protective with higher frequency (fA) reported among control (0.16) compared to the patients (0.13).

**Table 3.** Analytical comparison of the genotypes associated with myeloproliferative leukemia patients and healthy controls.

| Study Population | N= | CC % | CA % | AA % | C allele | A allele | DF | $\chi^2$ | *p* Value |
|---|---|---|---|---|---|---|---|---|---|
| | | | | *Bcl-2-938 C>A rs2279115* | | | | | |
| Patients | 100 | 34 (34.0%) | 44 (44.0%) | 22 (22.0%) | 0.56 | 0.44 | 2 | 7.32 | 0.025 |
| Healthy controls | 100 | 49 (49.0%) | 41 (41.0%) | 10 (10.0%) | 0.67 | 0.31 | | | |
| | N= | GG % | GA % | AA % | G allele | A allele | DF | $\chi^2$ | *p* value |
| | | | | *Bax-248G>A* | | | | | |
| Patients | 108 | 56 (51.85%) | 45 (41.66%) | 07 (6.48%) | 0.73 | 0.27 | 2 | 6.23 | 0.048 |
| Healthy controls | 110 | 75 (68.50%) | 30 (27.27%) | 5 (4.54%) | 0.82 | 0.18 | | | |
| Subjects | N= | TT | AT | AA | T | A | Df | $\chi^2$ | *p* value |
| | | | | *IL-8 rs4073 T>A* | | | | | |
| Patients | 113 | 19 (16.81%) | 41 (36.28%) | 53 (56.90%) | 0.25 | 0.75 | 2 | 20.24 | 0.0001 |
| Healthy controls | 122 | 48 (39.34%) | 46 (37.70%) | 28 (22.95%) | 0.39 | 61 | | | |
| Subjects | N= | GG | GA | AA | G | A | Df | $\chi^2$ | *p* value |
| | | | | *TNF-α rs1800629 G>A* | | | | | |
| Patients | 113 | 94 (83.18%) | 10 (8.84%) | 09 (7.96%) | 0.87 | 0.13 | 2 | 7.64 | 0.021 |
| Healthy controls | 122 | 100 (69.67%) | 20 (28.68%) | 02 (1.63%) | 0.84 | 0.16 | | | |

## 3.8. Logistic Regression Analysis of Bcl-2-938 C>A in Leukemia Patients

Table 4 presents the results from an unconditional logistic regression that aimed to investigate the link between genotypes and the risk of myeloproliferative leukemia. According to the analysis, there was a correlation between the Bcl-2-AA genotype and a higher likelihood of leukemia. The extent of this risk depended on the number of alleles present. In terms of the codominant model, individuals with the AA genotype of the Bcl-2 gene had a greater vulnerability to leukemia with an odds ratio (OR) of 3.17 (95% confidence interval (CI): 1.33–7.53), a relative risk (RR) of 1.88 (95% CI: 1.09–3.25), with a significant *p*-value (<0.009). The dominant model (CA+AA) showed a significant association between this genotype and susceptibility to leukemia with an OR of 1.86 (95% CI: 1.05–3.29), RR of

1.35 (95% CI: 1.030–1.77), and *p*-value of <0.03. Furthermore, the recessive model (CC+CA) compared to the AA genotype suggested an increased susceptibility to leukemia with an OR of 2.53 (95% CI: 1.13–5.68), RR of 1.71 (95% CI: 1.006–2.92), and *p*-value of <0.02. Finally, the A allele was linked to an increased risk of leukemia with an OR of 1.79 (95% CI: 1.18–2.69), RR of 135 (95% CI: 1.08–1.68), and *p*-value of <0.007.

**Table 4.** Association of *Bcl-2-938 C>A* genotypes with the risk of leukemia (Multivariate analysis).

| Genotypes | Controls (N = 100) | Patients (N = 100) | Odd Ratio (95% CI) | Risk Ratio (95% CI) | *p* Value |
|---|---|---|---|---|---|
| Codominant | | | | | |
| *Bcl-2 (CC)* | 49 (49.0%) | 34 (34.0%) | (ref.) | (ref.) | |
| *Bcl-2 (CA)* | 41 (41.0%) | 44 (44.0%) | 1.540 (0.840–2.840) | 1.220 (0.920–1.620) | 0.160 |
| *Bcl-2 (AA)* | 10 (10.0%) | 22 (49.0%) | 3.170 (1.330–7.530) | 1.880 (1.090–3.250) | 0.009 |
| Dominant | | | | | |
| *Bcl-2 (CC)* | 49 (49.0%) | 34 (34.0%) | (ref.) | (ref.) | |
| *Bcl-2 (CA + AA)* | 51 (51.0%) | 66 (66.0%) | 1.860 (1.050–3.290) | 1.350 (1.030–1.770) | 0.030 |
| Recessive | | | | | |
| *Bcl-2 (CC + CA)* | 90 (90.0%) | 78 (78.0%) | (ref.) | (ref.) | |
| *Bcl-2 (AA)* | 10 (10.0%) | 22 (22.0%) | 2.530 (1.130–5.680) | 1.710 (1.006–2.920) | 0.020 |
| Allele | | | | | |
| *Bcl-2 (C)* | 139 (69.50%) | 112 (56.0%) | (ref.) | (ref.) | |
| *Bcl-2 (A)* | 61 (30.50%) | 88 (44.0%) | 1.790 (1.180–2.690) | 1.350 (1.08–1.68) | 0.005 |

*3.9. Logistic Regression Analysis of Bax-248 G>A in Leukemia Patients*

To estimate the relationship between genotypes and myeloproliferative leukemia risk, an unconditional logistic regression was employed and presented in Table 5. The results showed that the genotype GG for *Bax-248 G>A* SNP has a significant association to an enhanced risk of leukemia. In the codominant model, the risk was observed with an odds ratio (OR) of 2.0 (1.12–3.57), a relative risk (RR) of 1.43 (1.045–1.95) and *p*-value of less than 0.017. Similarly, the dominant model (GA+AA vs. GG) was also related to a higher degree of risk to leukemia disease, with an OR of 1.98 (1.14–3.45), RR of 1.43 (1.05–1.91) and *p*-value of less than 0.014. However, no significant association was detected in the recessive model (GG+GA vs. AA). Our findings indicate that the allele A for *Bax-248 G>A* SNP has an association with a greater risk of leukemia, with an OR of 1.63 (1.03–2.58), RR of 1.29 (1.00–1.67), and a *p* value of less than 0.035.

**Table 5.** Association of *Bax-248G>A* genotypes with the risk of leukemia (Multivariate analysis).

| Genotypes Inheritance Model | Controls (N = 110) | % | Patients (N = 108) | % | Odd Ratio (95% CI) | Risk Ratio (95% CI) | *p* Value |
|---|---|---|---|---|---|---|---|
| Codominant | | | | | | | |
| *Bax (GG)* | 75 | 68.18% | 56 | 51.85% | (ref.) | (ref.) | |
| *Bax (GA)* | 30 | 27.27% | 45 | 41.66% | 2.0 (1.12–3.57) | 1.43 (1.045–1.95) | 0.01 |
| *Bax (AA)* | 05 | 4.54% | 07 | 4.68% | 1.87 (0.56–6.21) | 1.37 (0.69–2.72) | 0.30 |
| Dominant | | | | | | | |
| *Bax (GG)* | 75 | 68.18 | 56 | 51.85 % | (ref.) | (ref.) | |
| *Bax (GA + AA)* | 35 | 31.82 | 52 | 48.15 % | 1.98 (1.14–3.45) | 1.43 (1.05–1.91) | 0. 01 |
| Recessive | | | | | | | |
| *Bax (GG + GA)* | 105 | 95.45 | 101 | 93.52% | (ref.) | (ref.) | |
| *Bax (AA)* | 05 | 4.55 | 07 | 6.48 % | 1.45 (0.44–4.73) | 1.23 (0.61–2.42) | 0.53 |
| Allele | | | | | | | |
| *Bax (G)* | 180 | 81.82 | 157 | 73.36% | (ref.) | (ref.) | |
| *Bax (A)* | 40 | 18.18 | 57 | 26.63% | 1.63 (1.03–2.58) | 1.29 (1.00–1.67) | 0.035 |

### 3.10. Logistic Regression Analysis of IL-8 rs4073 T>A Genotypes to Predict the Risk of Leukemia

Our research found a strong correlation between the IL-8 rs4073 TT genotype and leukemia vulnerability in the codominant model, with an OR of 2.25 (95%) CI = (1.1433 to 4.434), RR = 1.35 (95%) CI = (1.0562 to 1.738) and $p < 0.018$ (Table 6). In the codominant inheritance model, there was a robust association between the IL-8rs4073 TA genotypes and the risk of leukemia, with an OR = 4.7 (95%) CI (2.379 to 9.64), RR = 2.07 (95%) CI (1.48 to 2.898) and $p < 0.0001$. The dominant inheritance model also showed a strong correlation between the IL-8 rs4073 TT and (TA+AA) genotypes and leukemia susceptibility, with an OR = 3.20 (95%) CI (1.73 to 5.92), RR= 1.62 (95%) CI 1.29 to 2.04) and $p < 0.0002$. Additionally, an intriguing finding was the correlation between the IL-8 rs4073 AA and (TT+TA) genotypes and leukemia vulnerability with an OR = 2.96 (95%) CI (1.6953 to 5.112), RR = 1.76 (95%) CI (1.276 to 2.447) and $p < 0.0001$ in the recessive inheritance model. In the allelic comparison, the IL-8 rs4073 -A allele was associated with leukemia susceptibility with an OR of 2.59, (95% CI) (1.7857 to 3.767), RR 1.56, and $p < 0.001$ (Table 6).

**Table 6.** Association of *IL-8 rs4073 T>A* genotypes with the risk of leukemia (Multivariate analysis).

| Genotypes Inheritance Model | Controls (N = 122) | Patients (N = 113) | Odd Ratio (95% CI) | Risk Ratio (95% CI) | *p* Value |
|---|---|---|---|---|---|
| Codominant | | | | | |
| *IL-8 (TT)* | 48 | 19 | (ref.) | (ref.) | |
| *IL-8 (TA)* | 46 | 41 | 2.25 (1.1433 to 4.434) | 1.35 (1.0562 to 1.738) | 0.018 |
| *IL-8 (AA)* | 28 | 53 | 4.7 (2.3714 to 9.642) | 2.07 (1.4820 to 2.898) | 0.0001 |
| Dominant | | | | | |
| *IL-8 (TT)* | 48 | 19 | (ref.) | (ref.) | |
| *IL-8-(TA + AA)* | 74 | 94 | 3.20 (1.7395 to 5.920) | 1.62 (1.2956 to 2.041) | 0.0002 |
| Recessive | | | | | |
| *IL-8 (TT + TA)* | 94 | 60 | (ref.) | (ref.) | |
| *IL-8- (AA)* | 28 | 53 | 2.96 (1.6953 to 5.112) | 1.76 (1.2706 to 2.447) | 0.0001 |
| Allele | | | | | |
| *IL-8 (T)* | 142 | 79 | (ref.) | (ref.) | |
| *IL-8-A* | 102 | 147 | 2.59 (1.78057 to 3.7657) | 1.56 (1.314 to 1.871) | 0.0001 |
| Over Dominant | | | | | |
| *IL-8 (TT + AA)* | 76 | 72 | (ref.) | (ref.) | |
| *IL-8 (TA)* | 46 | 41 | 0.94 (0.5536 to 1.5924) | 0.97 (0.758 to 1.2723) | 0.82 |

### 3.11. Logistic Regression Analysis of TNF-α rs1800629 G>A Genotypes to Predict the Risk of Leukemia

Our examination demonstrated a significant linkage between the TNF-α-AA genetic makeup and the susceptibility to myeloproliferative leukemia under the codominant model. The odds ratio was 4.78 (95% CI: 1.008–22.73), the relative risk was 2.83 (95% CI: 0.803–10.005), and the *p*-value was less than 0.048 (as shown in Table 7). We also observed a strong correlation between the TNF-α rs1800629 AA and TNF-α-(GA+GG) genotypes with leukemia susceptibility in the recessive inheritance model. The odds ratio was 5.14 (95% CI: 1.08–24.33), the relative risk was 2.93 (95% CI: 0.8036–10.334), and the *p*-value was less than 0.038. In comparison, the TNF-α-A allele was not significantly related to leukemia susceptibility in allelic comparison, with an odds ratio of 1.33 (95% CI: 0.756–2.37), a relative risk of 1.15, and a *p*-value of less than 0.32.

**Table 7.** Association of *TNF-α rs1800629 G>A* genotypes with the risk of leukemia (Multivariate analysis).

| Genotypes<br>Inheritance Model | Controls<br>(N = 122) | Patients<br>(N = 113) | Odd Ratio (95% CI) | Risk Ratio (95% CI) | *p* Value |
|---|---|---|---|---|---|
| **Cododminant** | | | | | |
| *TNF-α (GG)* | 100 | 94 | (ref.) | (ref.) | |
| *TNF-α (GA)* | 20 | 11 | 0.58 (0.2661 to 1.286) | 0.79 (0.5951 to 1.072) | 0.182 |
| *TNF-α (AA)* | 02 | 08 | 4.78 (1.0081 to 22.732) | 2.83 (0.8034 to 10.005) | 0.048 |
| **Dominant** | | | | | |
| *TNF-α (GG)* | 100 | 94 | (ref.) | (ref.) | |
| *TNF-α (GA+AA)* | 22 | 19 | 0.96 (0.4960 to 1.885) | 0.98 (0.7153 to 1.353) | 0.92 |
| **Recessive** | | | | | |
| *TNF-α (GA+GG)* | 120 | 105 | (ref.) | (ref.) | |
| *TNF-α (AA)* | 02 | 09 | 5.14 (1.0867 to 24.337) | 2.93 (0.8324 to 10.336) | 0.038 |
| **Allele** | | | | | |
| *TNF-α (G)* | 220 | 199 | (ref.) | (ref.) | |
| *TNF-α (A)* | 24 | 29 | 1.33 (0.7526 to 2.371) | 1.15 (0.8507 to 1.580) | 0.32 |
| **Over dominant** | | | | | |
| *TNF-α-(GG+AA)* | 102 | 105 | (ref.) | (ref.) | |
| *TNF-α (GA)* | 20 | 08 | 0.53 (0.2438 to 1.170) | 0.76 (0.5684 to 1.026) | 0.11 |

*3.12. Genotypes Association with Clinico-Pathological Features of the Leukemia Patients*

Table 8 demonstrates that the study identified a significant variance in the Bcl-2-938 C>A genotypes for different stages of CML, notably between chronic and accelerated phases. Nonetheless, there were no substantial differences found in the frequency of genotypes in relation to age and gender. Likewise, the correlation between Bcl-2-938 C>A genotypes and various types of MPD proved to be insignificant. Patients in CP revealed a higher degree of heterozygosity of CA genotype (60%) than the AP (25%) and BC phase (46%). The AP phase had a greater frequency of AA genotype compared to the CP (25%) and BC phase (15%). In the BC phase, the frequency of CC genotype was shown to be 38.46% in contrast to the CP-phase (20%) and AP-Phase (50%). Notably, there was a statistically significant variance in the frequencies of Bcl-2-938 C>A genotypes between CP-Phase and AP-Phase ($p < 0.04$). The analysis also indicates a significant difference in the frequencies of Bax-248 G>A genotypes for the age of the leukemia patients, with a higher frequency of the AA genotype among older patients. However, there was no significant correlation with gender, CML stages or different types of MPD. A greater frequency of the AA genotype was also observed in the blastic phase. The TNF-α rs1800629 G>A genotypes showed a significant difference between the CP-CML vs. BC-CML but a non-significant difference between CP-CML and AP-CML. Significant differences were observed in the frequencies of TNF-α rs1800629 G>A genotypes to age and gender. No significant correlation was reported between TNF-α rs1800629 genotypes and MPD groups. The study also found a significant difference in the frequencies of IL-8 rs4073 T>A genotypes to age and gender, but no significant correlation with CML stages or other MPDs. The comparative analysis of different genotypes did not reveal any significant differences between Ph+ and Ph− patients (Table 9). It is important to note that while Ph−chromosome positivity is important in leukemias, there are different Ph transcripts with distinct clinical and genetic associations. This can make the Ph+ and Ph− groups of patients even more heterogeneous in terms of interpreting results.

**Table 8.** Association of genotypes with clinical characteristic of the MPDs.

| | | | | *Bcl-2-938 C>A* genotypes | | | | |
|---|---|---|---|---|---|---|---|---|
| | Gender | N = 100 | CC % | CA % | (AA) % | Df | X² | *p* Value |
| Age | >40 | 60 | 20 (33.33%) | 30 (50%) | 10 (16.66%) | 2 | 3.19 | 0.20 |
| | <40 | 40 | 14 (35%) | 14 (35%) | 12 (30%) | | | |
| Gender | Males | 78 | 28 (35.89%) | 36 (46.15%) | 14 (17.95%) | 2 | 4.89 | 0.08 |
| | Females | 22 | 06 (27.27%) | 08 (36.36%) | 08 (36.36%) | | | |
| Stage | CP-CML | 25 | 5 (20%) | 15 (60%) | 5 (20%) | 2 | 5.43 | 0.04 |
| | AP-CML | 20 | 10 (50%) | 5 (25%) | 5 (25%) | | | |
| | CP-CML | 25 | 5 (20%) | 15 (60%) | 5 (20%) | 2 | 1.5 | 0.47 |
| | BC-CML | 13 | 5 (38.46%) | 6 (46.15%) | 2 (15.38%) | | | |
| MPD types | MPD ET | 15 | 6 (0.4%) | 5 (33.33%) | 4 (26.67%) | 0.64 | 4 | 0.95 |
| | MPD PV | 12 | 5 (41.67%) | 5 (41.67%) | 2 (16.67%) | | | |
| | MPD MF | 06 | 3 (0.5%) | 2 (33.33%) | 1 (16.67%) | | | |
| | | | | *Bax-248G>A* genotypes | | | | |
| | | N = 108 | GG (56) | GA (45) | AA (07) | Df | X² | *p* value |
| Age | >40 | 63 | 26 (41.26%) | 35 (55.55%) | 02 (3.17%) | 2 | 12.82 | 0.001 |
| | <40 | 45 | 30 (66.66%) | 10 (22.22%) | 05 (11.11%) | | | |
| Gender | Males | 73 | 36 (49.32%) | 33 (45.21%) | 03 (4.11%) | 2 | 2.83 | 0.24 |
| | Females | 35 | 20 (57.14%) | 12 (34.29%) | 04 (11.43%) | | | |
| Stage | CP-CML | 30 | 14 (46.67%) | 13 (43.33%) | 03 (1%) | 2 | 0.31 | 0.98 |
| | AP-CML | 14 | 06 (42.86%) | 07 (0.5%) | 01 (7.14%) | | | |
| | CP-CML | 30 | 14 (46.67%) | 13 (43.33%) | 03 (1%) | 2 | 0.15 | 0.92 |
| | BC-CML | 10 | 04 (40%) | 05 (50%) | 01 (10%) | | | |
| MPD types | MPD ET | 30 | 18 (6%) | 11 (36.67%) | 1 (3.33%) | 4 | 0.80 | 0.93 |
| | MPD PV | 16 | 10 (62.5%) | 05 (31.25%) | 1 (6.25%) | | | |
| | MPD MF | 04 | 02 (0.5%) | 2 (0.5%) | 0 (0%) | | | |
| | | | | *TNF-α rs1800629 G>A* genotypes | | | | |
| | | N = 113 | GG94 | GA10 | AA09 | Df | X² | *p* value |
| Age | >40 | 76 | 70 (92.10%) | 02 (3.94%) | 04 (5.26%) | 14.49 | 2 | 0.0007 |
| | <40 | 37 | 24 (64.86%) | 8 (21.62%) | 05 (13.51%) | | | |
| Gender | Males | 82 | 72 (87.80%) | 08 (9.75%) | 02 (3.65%) | 12.5 | 2 | 0.0019 |
| | Females | 31 | 22 (70.96%) | 02 (9.67%) | 07 (19.35%) | | | |
| Stage | CP-CML | 34 | 27 (79.41%) | 04 (11.76%) | 03 (8.82%) | 1.04 | 2 | 0.59 |
| | AP-CML | 20 | 18 (90%) | 01 (5%) | 01 (5%) | | | |
| | CP-CML | 34 | 27 (79.41%) | 04 (11.76%) | 03 (8.82%) | 7.32 | 2 | 0.025 |
| | BC-CML | 10 | 5 (50%) | 05 (50%) | 00 (0%) | | | |
| MPD types | MPD ET | 30 | 24 (80%) | 3 (10%) | 03 (10%) | 4.56 | 4 | 0.332 |
| | MPD PV | 15 | 11 (73.33%) | 3 (20%) | 1 (6.66%) | | | |
| | MPD MF | 04 | 2 (50%) | 2 (50%) | 0 (0%) | | | |
| | | | | *IL-8 rs4073 T>A* genotypes | | | | |
| | | N = 113 | TT19 | TA41 | AA53 | Df | X² | *p* value |
| Age | >40 | 76 | 08 (10.52%) | 31 (40.78%) | 37 (48.68%) | 6.91 | 2 | 0.031 |
| | <40 | 37 | 11 (29.72%) | 10 (27%) | 16 (43.24%) | | | |
| Gender | Males | 82 | 8 (9.75%) | 33 (40.24%) | 41 (50%) | 10.76 | 2 | 0.004 |
| | Females | 31 | 11 (35.48%) | 8 (25.80%) | 12 (38.70%) | | | |
| Stage | CP-CML | 34 | 06 (17.64%) | 12 (35.29%) | 16 (47.05%) | 2 | 0.32 | 0.85 |
| | AP-CML | 20 | 03 (15%) | 06 (30%) | 11 (55%) | | | |
| | CP-CML | 34 | 06 (17.64%) | 12 (35.29%) | 16 (47.05%) | 2 | 4.32 | 0.11 |
| | BC-CML | 10 | 05 (50%) | 02 (20%) | 03 (30%) | | | |
| MPD types | MPD ET | 30 | 02 (6.66%) | 16 (53.33%) | 12 (40%) | 4.21 | 4 | 0.37 |
| | MPD PV | 15 | 02 (13.33%) | 04 (26.66%) | 9 (60%) | | | |
| | MPD MF | 04 | 01 (25%) | 01 (25%) | 02 (50%) | | | |

**Table 9.** Comparative association of the gene polymorphisms between Ph+ and Ph− cases of MPD.

| Association of *Bcl-2-938* gene polymorphism | | | | | | | |
|---|---|---|---|---|---|---|---|
| | N = 91 | CC(34) | CA (38) | AA (19) | Df | $\chi^2$ | *p* value |
| Ph (+) patients | 58 | 20 | 26 | 12 | 2 | 0.28 | 0.86 |
| Ph (−) patients | 33 | 14 | 12 | 07 | | | |
| Association of *Bax-248G>A* gene polymorphism | | | | | | | |
| | N = 94 | GG(44) | GA(43) | AA(07) | Df | $\chi^2$ | *p* value |
| Ph (+) patients | 54 | 24 | 25 | 05 | 2 | 0.72 | 0.69 |
| Ph (−) patients | 40 | 20 | 18 | 02 | | | |
| Association of *TNF-α rs1800629 G>A* polymorphism | | | | | | | |
| | N = 113 | GG87 | GA18 | AA08 | Df | $\chi^2$ | *p* value |
| Ph (+) patients | 64 | 50 | 10 | 04 | 2 | 0.18 | 0.91 |
| Ph (−) patients | 49 | 37 | 08 | 04 | | | |
| Association of *IL-8 rs4073 T>A* gene polymorphism | | | | | | | |
| | N = 112 | TT19 | TA41 | AA53 | Df | $\chi^2$ | *p* value |
| Ph (+) patients | 63 | 13 | 20 | 30 | 2 | 2.8 | 0.24 |
| Ph (−) patients | 49 | 05 | 21 | 23 | | | |

## 4. Discussion

According to the WHO classification of the hematopoietic neoplasm, the major myeloid subclasses include chronic myeloproliferative disorders (MPDs) which represent a heterogeneous group of neoplastic diseases that could be subdivided based on the presence of the Philadelphia chromosome (Ph+) such as CML and others which are Philadelphia chromosome negative (Ph−) myeloproliferative disorders which include essential thrombocytopenia, polycythemia verra and myelofibrosis [44]. It is observed that different phases of CML have varying degree of prognosis as patients progressing from the chronic phase (which responds well to TKI therapy) to the accelerated phase and blast crisis have shown poor clinical outcomes [45]. Studies have proposed that the genetic drivers in the myeloid cell lineage are significant in influencing the progression of CML to advanced clinical stages [46,47]. These myeloproliferative neoplasms represent a heterogeneous group which can present in various disease phenotypes and have the potential to transform into one another [48]. The non-CML MPDs are considered to be genetically heterogeneous though these are believed to be related as the aberrant clone arising from a progenitor which is a multipotent hematopoietic stem cell [49]. It is in the subsequent clonal expansions that accumulation of genetic alterations results in committing the cells towards a specific MPD. The most frequent type of genetic variation found in the human genome is Single Nucleotide Polymorphisms (SNPs). This refers to alterations in a single base pair of a nucleotide that varies from one individual to another. Significant functional effects can arise from SNPs found in the promoter regions of genes, as these regions regulate gene expression. These specific SNPs can act as markers for diagnosing and predicting the prognosis of cancer, as they influence gene function and expression, ultimately impacting disease susceptibility and responsiveness to treatment. Gene regulations causing the quantitative and qualitative variations in the product can be greatly influenced by variations in the allele in the promoter region of target genes which might alter the transcription factor-binding site. A multitude of SNPs have been implicated in carcinogenesis and these have been shown to effectively influence cellular mechanisms such as apoptosis which is inherently utilized to get rid of cells with the potential to develop into cancer [50,51].

Apoptosis refers to a process whereby cells undergo programmed cell death, aiming to get rid of unwanted or unhealthy cells in the body. However, the effect inflammation has on apoptosis can either be positive or negative depending on the circumstances. In some scenarios, inflammation can aid apoptosis by eliminating damaged cells caused by infections or acute injuries. On the other hand, chronic inflammation can hinder

apoptosis, resulting in the accumulation of weird or abnormal cells in the body, increasing the risk of cancer and other ailments. Inflammation thwarts apoptosis by stimulating signaling pathways that encourage cell growth and survival, thus preventing cell death. This may lead to the persistence of unusual cells that should have undergone apoptosis, promoting the growth of cancer cells. All in all, it is apparent that the correlation between inflammation and apoptosis is intricate and also contingent on the conditions. BCL-2 protein has numerous roles in cellular pathways that are related to cell death or cell survival as well as act as important factors in resistance to cancer therapeutics. There are instances where polymorphisms in the *Bcl-2* gene have been shown to alter its anti-apoptotic role thereby promoting cell death and aiding in targeted therapies [25,52]. Our study examined the P2 promoter polymorphisms of the *Bcl-2* gene. This functions as a negative regulator element in the process. There was no significant difference in the genotype and allelic frequencies of the *Bcl-2-938 C>A* (rs2279115) polymorphism between healthy controls and PV or ET. In case of CML, *Bcl-2* gene polymorphic variation with control was significant. Since the expression of the gene is elevated in CML, it has been earlier proposed as a marker in post-chemotherapy prognosis of myeloid leukemia. It has been observed that the *Bcl-2-938 C>A* gene variation is statistically related to overall survival and remission rates in leukemia [15]. Moreover, a study also reported a significant correlation between the *Bcl-2-938* gene polymorphism and the susceptibility to CML disease [53]. The pro-apoptotic roles of the *Bax* gene to other collaborative genes, such as *Bcl-2* and *p53*, have renewed the interest in Bax gene modifications and the related outcome in cancer. *Bax* has been widely researched on numerous types of cancers, including pancreatic cancer [54], colon cancer [55,56], esophageal cancer [57,58], pulmonary cancer [59,60], squamous head and neck cancer [61], prostate cancer [62], ovarian cancer [63], and breast cancer [64]. The *Bax* gene has been involved in the suppression of tumors due to its role in promoting the programmed cell death. The experimental deletions of the *Bax* gene have been shown to be associated with the incidence of lymphoid hyperplasia and thus it is regarded as an important suppressor against hematopoietic neoplasms [65–68]. There was no significant difference in the genotype and allelic frequencies of *Bax-248 G>A* (rs4645878) polymorphisms between healthy controls and PV or ET patients. This finding is consistent with studies on solid cancers, as reported by Alam et al. in 2019 [69]. Although *Bax rs4645878* does not seem to play a role in carcinogenesis, it has been suggested that it may be associated with a poor prognosis in some cancers [70].

Higher serum levels of IL-8 have been seen in advanced stages and progression of various cancer types that include colorectal, breast cancer, prostate, pancreatic, melanoma, ovarian and renal cancer [71]. Studies in the literature demonstrate that most of the work has been conducted on the effect of proinflammatory cytokine IL-10 genomic variations on various cancers including hematological malignances [72,73]. However, the effect of *IL-8* polymorphic variations on the risk and susceptibility to various cancers, particularly leukemia, has been largely unexplored which gives credence to our attempt to characterize the *IL-8 rs4073 T>A* polymorphism in leukemia patients in comparison to healthy subjects. Interleukin-8 has been found to be overexpressed in breast cancer including HER2-positive cancers compared with normal breast tissue and has been shown to enhance breast cancer progression by promoting angiogenesis, cell invasion and distant colonization [74]. A meta-analysis of 33 studies, with 6192 cases in a systematic review, has recently reported the significance of *IL-8* genetic polymorphism *IL-8-251T>A* (rs4073) as a risk factor in gastric cancer which was stratified based on ethnicity with such an association observed in Korean, Chinese and Brazilian patients though not among Japanese patients [75].

A recent study, which addressed IL-8 as a pro-tumorigenic cytokine that promotes cancer proliferation and migration, found *IL-8 rs4073* AT and AA genotypes to have significantly lower prevalence in TNBC patients, while individuals with allele A showed a relatively decreased risk for breast cancer [71]. Our finding is contrary to this study where we have found that leukemia patients had a higher frequency of the A allele than healthy controls which makes it a risk allele for the disease. However, such a contradiction between

solid and hematological malignances needs further studies on the action mechanism of IL-8 in these cancer types as it is known to play a critical role in epithelial-mesenchymal transition for migration and metastasis of cancer cells in solid cancers [76], apart from the observed variations due to the ethnicity of the subjects under study. In patients (Mexican children) with neuroblastoma, the homozygous AA genotype of *rs4073 IL-8* appeared more frequently along with a high serum levels IL-8, which were associated with lower overall survival and the finding was further confirmed on the basis of an adverse prognosis in a multivariate model [77]. The pro-inflammatory cytokine TNF-α has originally been linked to necrosis in chemically induced sarcomas in mice and later with an ability to cause apoptosis of tumor cells through TNFR1 [78,79]. Over time, evidence also emerged to support some studies that showed TNF-α promoting cancer growth and progression [80]. The contrary roles could possibly be explained as a concentration-dependent phenomenon where a very high concentration of the cytokine acts as anti-tumor agent, whereas as in lower concentrations it helps the cancer cells to sustain, grow and proliferate. The cytokine has also been observed to be involved in inducing dedifferentiation processes leading to tumor relapse such as in case of a melanoma patient treated with ACT [81]. Moreover, the role of *TNF-α* has been well established in the metastatic pathway of EMT in various solid cancers including lung and breast cancer [82,83]. In the case of hematological malignances, TNF-α has been found to support the cellular microenvironment promoting progression of acute leukemia and its relapse. A recent study by Verma et al. [84] has demonstrated higher levels of the cytokine in T-ALL cases, which was followed by AML and B-ALL when compared to the control. However, the study observed a significant reduction in the serum TNF-α level in patients with acute leukemia with the start of chemotherapy [84]. A previous study on polymorphic variation of TNF–α rs1800629 (–308 G>A) in leukemia demonstrated that the frequency of the rs1800629 GA genotype was high in the patient group as compared to healthy controls and it was associated with high risk of adult B-ALL [85]. Similarly, another study found statistically significant associations of *TNF-α rs1800629* SNP in AML subtypes with a higher frequency of variant genotypes in AML de novo cases [86]. The *TNF-α AA* homozygous variant genotype also demonstrated significant risks of the development of pediatric ALL in a study though this association was not observed in adult ALL [87]. Our study provides an insight into the plausible roles of polymorphic gene variations of some important candidate genes in the risk and susceptibility to human myeloproliferative malignances. Moreover, such a genetic heterogeneity within a given population has been emerging as a novel genomic marker and a tool for personalized medicine for therapeutics as well as for prognosis of the disease.

## 5. Concluding Remarks

The data from the current study provide partial but important evidence that polymorphisms in apoptotic genes *Bcl-2-938C>A* and *Bax-248G>A* and pro-inflammatory cytokines *IL-8 rs4073 T>A* and *TNF-α rs1800629 G>A* may help predict the clinical outcomes of patients with leukemia. However, within the study the two major classes of MPD that included CML (Ph+) and Non-CML MPDs (Ph−) did not show any significant relationship when accounted for the distribution of the genotypes. The scope of MPDs is larger with regard to progression to the advanced stage as most of these have the potential to transform into acute myelogenous leukemia (AML). In fact, the diagnosis of CML-BC if not presented as a monitored prognosis in CML patients, possesses a substantial challenge in order to be differentiated from AML and might even mimic leukemia with lymphoid origin [88]. It is important to mention that certain genetic abnormalities drive the clonal evolution and blastic transformation of CML and the resulting resistance to TKIs therapy [89,90]. Our report has presented some interesting preliminary findings, which could be further validated in larger cohorts and diverse ethnic populations.

**Author Contributions:** Conceptualization, M.S.M., R.M. and M.F.U.; Methodology, R.M. and O.A.; Analysis, R.M.; M.F.U. and F.J.T.; Investigation, M.S.M.; O.A. and R.M., Supervision; R.M. and M.F.U., Project administration, F.J.T. and R.M., Writing, review and editing, R.M.; M.F.U., M.S.M. and F.J.T. All authors have read and agreed to the published version of the manuscript.

**Funding:** This work was supported by grants from the Deanship of Scientific Research (S-1440-0333), University of Tabuk. Saudi Arabia. We are highly thankful to the University of Tabuk and DSR for supporting our research study.

**Institutional Review Board Statement:** The study was conducted in accordance with the Declaration of Helsinki, and approved by the Ethics Committee of University of Tabuk (UT-88-19-2019 Dated 24 November 2019).

**Informed Consent Statement:** Informed consent was obtained from all subjects involved in the study.

**Data Availability Statement:** The data presented in this study are available within this open access article. Any further queries can be sent to the corresponding author.

**Conflicts of Interest:** The authors declare no conflict of interest.

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
