# Peer review of "Molecular Evaluation of the Impact of Polymorphic Variants in Apoptotic (Bcl-2/Bax) and Proinflammatory Cytokine (TNF-α/IL-8) Genes on the Susceptibility and Progression of Myeloproliferative Neoplasms: A Case-Control Biomarker Study"

_cimb, doi:10.3390/cimb45050251_

Round 1

Reviewer 1 Report (Previous Reviewer 2)

In revised version, the authors pointed to importance of Ph-chromosome positivity in leukemias. However, there are different Ph transcripts with distinct clinical/genetic associations. Therefore, this fact makes the Ph+ and Ph- groups of patients even more heterogenous in view of result interpretation. Moreover, the chronic myeloproliferative disorders, t.g., erythremia and thrombocytopenic purpura, may possess constitutive mutation of BRAF gene thus also influencing the outcomes in MPD patients.

Therefore, one may conclude only about a relatively genetically homogenous group of CML patients which, however, is rather small for proper statistical analysis of the mentioned Bcl-2 and Bax gene variants. Hence, the manuscript cannot be published in its revised form.

Author Response

Reviewer 2 Report (New Reviewer)

In the manuscript, Moawad et al. conducted a genetic investigation on single nucleotide polymorphisms (SNPs) of Bcl-2, Bax-2, IL-8, and TNF-α which are associated with the susceptibility to hematological malignancies, including CML and other myeloproliferative diseases (MPDs). In the outcome, the study recapitulated SNPs in these genes can be considered for molecular/genomic markers, as well as exploited for further precision medicines and disease prognosis. The rationale and significance of the study are appropriate. However, there are still some issues and concerns that the authors must address.

1. In the Abstract, the authors described many details in Results and Conclusions, making the Abstract too long. They should summarize and reduce the portion of the Results and Conclusions. By the way, they also should have a Simple Summary for the manuscript.

2. In the Introduction, again, the authors provided many unnecessary information which can be placed in the Discussion. Therefore, the Introduction should be reduced or switched the information to the Discussion.

3. Also in the Introduction, the authors should provide the link between apoptosis and proinflammation, in which SNPs of apoptosis-related genes (Bcl-2, Bax-2) and proinflammation-related genes (IL-8, TNF-α) can be studied simultaneously in one research.

4. The last two sentences in Introduction (marked in green color) look unnecessary in this study.

5. In the Sample Collection of Method, why did the authors only collect peripheral blood samples, and exclude bone marrow samples?

6. CBC, KFT, LFT should be written in full.

7. In the PCR gradient, how can the authors choose the indicated temperatures? They should put citations if it is based on previous studies.

8. The figure of agarose electrophoresis and descriptive context should be placed in Results. By the way, how can authors get PCR fragments in different sizes and how can they guarantee they are corresponding to indicated SNPs based on the size differences?

9. In the end of page 9, “TNF-α rs1800629” is missing G>A.

10. The two last sentences of the first paragraph in page 10, there is something wrong as frequency of A allele (fA) was mentioned twice.

11. The general legend of Table 3 may be missing as the current legend after Table 3 only goes to Bcl-2.

12. In the “Genotypes association with clinic-pathological features”, there is always a non-significant correlation between SNP genotypes and types of MPD, can the authors address the reason why the phenomenon occurs.

13. In Bcl-2 -938 C>A, it is weird when CML patients with BC-phase (the most aggressive phase of CML) have higher frequency of CC (50%). Can the authors clarify this result?

14. In the Author Contribution, the contribution of each author should be clearly described.

15. Minor: The authors should check again the entire manuscript for grammatical errors (typos, single/plural), consistency of gene/protein names etc. Gene names should be in Italic and the authors should remove “gene” after the gene name if the gene annotation has been Italic.

Round 2

Reviewer 1 Report (Previous Reviewer 2)

The manuscript is now sufficiently revised in response to the critical comments. The next version looks better arranged.

1. The rationals for studying pro-apoptotic gene variants are now explained.

2. Cytogenetic and clinical heterogeneity  (Ph+ vs Ph- patients etc.) is taken into account and discussed

3. The conclusions are more cautious, due to preliminary results of this small study in a heterogenous group of patients.

The article may be published in its current form.

Author Response

Reviewer 2 Report (New Reviewer)

Dear the authors,

Thank you very much for your efforts to revise the manuscript. Although the manuscript has been improved, there is still some issues that require your responses

Comment 2: The Introduction is still too long and complicated with many provided information, which look like the Discussion. The authors should provide the main information which can highlight the importance why you investigate SNPs of those genes.

Comment 3: Also in the Introduction, the additional explanation of the association between apoptosis and inflammation is too long. Most of the provided information can be transferred to the Discussion.

Comment 7: As group of the authors has already published several genotyping studies utilizing AMR-Primers and optimization of PCR gradient, they should add the citations in this method.

Comment 8: The Figure 2 and descriptive context have not been placed in Results. As they are the results, so keeping them in the Materials and Method is not appropriate.

Comment 13: As the explanation of the authors, in Bcl-2 -938 C>A, why is CC genotype (less leukemia susceptibility) favorite in both AP (later stage of CML) and BC phases (the most aggressive phase)? Those are even in higher frequency percentages than AA genotype (high leukemia susceptibility).

Minor: Gene names must be in Italic to distinguish to proteins.

Other comments have been fully addressed.

Round 3

Reviewer 2 Report (New Reviewer)

Dear authors,

I neither did not see nor receive the responses and revised manuscript from you to address my recent comments/concerns. Please refer this problem to Editors!

Author Response

This manuscript is a resubmission of an earlier submission. The following is a list of the peer review reports and author responses from that submission.

Round 1

Reviewer 1 Report

Dear authors,

Still, they are patients with myeloproliferative neoplasms, most in the chronic phase, and only 25% of these patients are in the accelerated/blastic phase, similar to acute leukemia. And patients with CLL are quite different patients.

You should better define the population and focus on myeloproliferative neoplasms in their different phases. And it would be interesting to compare the results between them, especially between Ph-positive and negative patients, for eg.

I think the idea of the article is quite interesting, but you need a better description of the cases and reorganization of the concept of “leukemia.” Mainly because you are talking about hematological neoplasm (most chronic) as a synonym for leukemia. 

Author Response

Dear Editor

Thank you for your email suggesting revision to our manuscript (CIMB-233005). We have revised the manuscript in the light of reviewers’ comments. The changes have been highlighted in red font. Further we have listed the responses to the comments.

We hope that the manuscript shall now be found suitable for publication.

Thank you,

Best regards!

Rashid Mir, PhD

Response:

Still, they are patients with myeloproliferative neoplasms, most in the chronic phase, and only 25% of these patients are in the accelerated/blastic phase, similar to acute leukemia. And patients with CLL are quite different patients.

We have modified the title replacing hematological cancers with myeloproliferative neoplasms. In addition, the data for CLL has been removed as suggested by the reviewers to focus just on the MPDs.

You should better define the population and focus on myeloproliferative neoplasms in their different phases. And it would be interesting to compare the results between them, especially between Ph-positive and negative patients, for eg.

The introduction has also been added with a paragraph to define MPDs and its heterogeneity for the better clarity as suggested by the reviewer.

I think the idea of the article is quite interesting, but you need a better description of the cases and reorganization of the concept of “leukemia.” Mainly because you are talking about hematological neoplasm (most chronic) as a synonym for leukemia. 

Manuscript has been revised to our best as per the reviewers comments.

Reviewer 2 Report

The article concerns potential modifying effects of previously described common BAX and BCL-2 promoter gene variants which could modulate apoptosis of hematopoietic cells in leukemia patients, thus influencing outcomes of the disease. The authors also attempted to evaluate association with susceptibility. The study is performed in rather small samples of leukemia patients using a case-control study approach (a total of 244 cases). Along with Bax and BCL-2 genes, the well-known TNF and IL-8 polymorphisms were studied. The authors have found some differences between controls and leukemia patients, e.g., for Bcl-2 -938 C>A, Bax-248G>A. Appropriate risk factors were evaluated in different statistical models. The PCR methodology is well described.

Remarks:

Statistical analysis (p.9): The assumed level of p-values (< 0.05) may be not too reliable for the difference assessment, especially in cases of small differences between the mean allelic frequencies in small subgroups .

The selected group of patients is quite heterogenous, it includes chronic myeloid leukemia (CML, 64 cases) as well as other chronic myeloproliferative conditions (N=54) as seen from Table 2. These malignancies originate from different steps of stem cell maturation, thus, probably, showing different gene-cell relations. This aspect should be considered in Discussion.

Numerous results presented in the article, generally, reflect small but significant case/control differences in BAX and BCL-2 allelic variants, and quite interesting interrelations with TNF-α and IL-8 polymorphisms (p.13). However, the relatively small number of cases at different clinical stages of CML (page 15) are insufficient to make reliable statistics. Hence, the section on Genotypes association with clinico-pathological features may be abridged or even skipped, despite high p values for some factors (p.16, Table 8).

In Conclusion one should clearly suggest which gene variants affect susceptibility to leukemia, and what polymorphisms influence clinical outcomes of the disease. Moreover, small group of patients/controls should be also a limiting factor for proper assessment of results. Therefore, the paper could be published after some revision, but should be considered a preliminary report.

Author Response

Dear Editor

Thank you for your email suggesting revision to our manuscript (CIMB-233005). We have revised the manuscript in the light of reviewers’ comments. The changes have been highlighted in red font. Further we have listed the responses to the comments.

We hope that the manuscript shall now be found suitable for publication.

Thank you,

Best regards!

Rashid Mir, PhD

Response:

Statistical analysis (p.9): The assumed level of p-values (< 0.05) may be not too reliable for the difference assessment, especially in cases of small differences between the mean allelic frequencies in small subgroups .

We have modified the title replacing hematological cancers with myeloproliferative neoplasms. In addition, the data for CLL has been removed as suggested by the reviewers to focus just on the MPDs.

The selected group of patients is quite heterogenous, it includes chronic myeloid leukemia (CML, 64 cases) as well as other chronic myeloproliferative conditions (N=54) as seen from Table 2. These malignancies originate from different steps of stem cell maturation, thus, probably, showing different gene-cell relations. This aspect should be considered in Discussion.

The table has also been modified to reflect Ph+ and Ph- MPDs and their risk associated genotypes.

Numerous results presented in the article, generally, reflect small but significant case/control differences in BAX and BCL-2 allelic variants, and quite interesting interrelations with TNF-α and IL-8 polymorphisms (p.13). However, the relatively small number of cases at different clinical stages of CML (page 15) are insufficient to make reliable statistics. Hence, the section on Genotypes association with clinico-pathological features may be abridged or even skipped, despite high p values for some factors (p.16, Table 8).

With regard to table 8, we believe that the data shows some interesting findings and we request the reviewer to let us have this table in the manuscript as clinicopathological associations with some genotypes does reflect a significant piece of scientific data for biomarker studies.

In Conclusion one should clearly suggest which gene variants affect susceptibility to leukemia, and what polymorphisms influence clinical outcomes of the disease. Moreover, small group of patients/controls should be also a limiting factor for proper assessment of results. Therefore, the paper could be published after some revision, but should be considered a preliminary report.

  • Thankyou very much for reviewing our manuscript. The conclusion has been revised as per our results .
